# Wi-Fi-Based Indoor Localization and Navigation: A Robot-Aided Hybrid Deep Learning Approach

**DOI:** 10.3390/s23146320

**Published:** 2023-07-12

**Authors:** Xuxin Lin, Jianwen Gan, Chaohao Jiang, Shuai Xue, Yanyan Liang

**Affiliations:** 1Faculty of Innovation Engineering, Macau University of Science and Technology, Macau 999078, China; linxuxin6@gmail.com (X.L.); jackyganchina@hotmail.com (J.G.); 2009853zii30002@student.must.edu.mo (C.J.); 1909853gii30010@student.must.edu.mo (S.X.); 2Zhuhai Da Heng Qin Technology Development Co., Ltd., Zhuhai 519000, China; 3School of Applied Science and Civil Engineering, Beijing Institute of Technology Zhuhai, Zhuhai 519000, China

**Keywords:** Wi-Fi-based indoor localization and navigation, deep reinforcement learning, semi-supervised learning, unsupervised learning

## Abstract

Indoor localization and navigation have become an increasingly important problem in both industry and academia with the widespread use of mobile smart devices and the development of network techniques. The Wi-Fi-based technology shows great potential for applications due to the ubiquitous Wi-Fi infrastructure in public indoor environments. Most existing approaches use trilateration or machine learning methods to predict locations from a set of annotated Wi-Fi observations. However, annotated data are not always readily available. In this paper, we propose a robot-aided data collection strategy to obtain the limited but high-quality labeled data and a large amount of unlabeled data. Furthermore, we design two deep learning models based on a variational autoencoder for the localization and navigation tasks, respectively. To make full use of the collected data, a hybrid learning approach is developed to train the models by combining supervised, unsupervised and semi-supervised learning strategies. Extensive experiments suggest that our approach enables the models to learn effective knowledge from unlabeled data with incremental improvements, and it can achieve promising localization and navigation performance in a complex indoor environment with obstacles.

## 1. Introduction

With the rapid development of wireless technologies, location-based services have received increasing attention in practical applications by providing the localization and navigation functions in outdoor or indoor environments. Although the global positioning system (GPS)-based technology has achieved high accuracy for outdoor location identification, it cannot be naturally applied to the indoor localization task due to weak signals and complex environments with various obstacles [1].

Over the past decade, many technologies have been investigated for indoor localization and navigation using different wireless networks, such as Wi-Fi [2], Bluetooth [3], Radio Frequency Identification Device (RFID) [4] and Ultra Wide Band (UWB) [5]. Among these technologies, the Wi-Fi-based technology has a greater deployment potential due to the widespread distribution of Wi-Fi infrastructure in indoor public spaces, such as museums, shopping malls, and large parking lots.

Many existing Wi-Fi-based methods have been proposed to solve the indoor localization problem by using trilateration [6] or fingerprint identification techniques [7]. The methods using trilateration require the exact locations of the Wi-Fi access points (APs), which are difficult to obtain in some specific environments such as shopping malls. In contrast, the fingerprint-based methods only need to collect the received signal strength indicator (RSSI) data for known indoor positions to create an offline radio map with the measured data and corresponding location coordinates. During the online phase, the real-time position of the target device could be estimated by using the current measured RSSI data and the radio map. However, the process of creating and maintaining such a Wi-Fi radio map is very time-consuming, as the RSSI data usually need to be densely labeled with accurate coordinates for reliable position estimation. In addition, hardware differences can have a significant impact on the localization accuracy when different devices, such as mobile phones, are used to collect Wi-Fi signals [8]. Some other factors, such as physical barriers, the presence of people and multi-path effects, are also needed to be taken into account as they may lead to the gradual attenuation of Wi-Fi signals.

In recent years, many machine learning (ML) algorithms [9,10,11,12,13,14,15,16,17,18,19] have been applied to the fingerprint-based positioning system. They could learn useful knowledge from multi-dimensional measured data with position labels to reduce the effect of RSSI fluctuation and improve fingerprinting accuracy and system robustness. However, these methods still require a large amount of high-quality annotated data in the data collection phase to handle various unknown environments. Traditional manual annotation requires participants to keep moving and collect the RSSI data while recording their current position. This type of data collection is difficult and unreliable because participants are easily distracted when collecting data repeatedly in a large indoor space. In addition, most of the existing methods do not further consider the use of unlabeled data during model training, which is easier to obtain than labeled data in the data collection phase.

To alleviate the above problems, we need to consider two key issues: (1) how to build a unified and automatic data collection scheme to enhance the quality of offline data and (2) how to make full use of unlabeled data to improve online indoor localization and navigation accuracy. In this paper, we propose a robot-aided hybrid deep learning approach, called RA-HDL. Figure 1 shows the overall workflow of our approach, including data collection, model training and model deployment. Our main contributions are summarized as follows:

•We utilize a robot to automatically collect data in an indoor environment with obstacles. The data include a large amount of unlabeled Wi-Fi RSSI data and a small amount of Wi-Fi RSSI data labeled by grid coordinates. The data can be used to train and test the models in our experiments and will be made publicly available.•We design two deep learning models for the Wi-Fi-based localization and navigation tasks, respectively. Both models are constructed based on a densely fused variational autoencoder (DF-VAE) that could learn the robust latent feature representation and improve the gradient flow when training a deep network.•Based on the above data and models, we develop a novel hybrid learning approach consisting of supervised, unsupervised and semi-supervised learning strategies. This approach could take full advantage of the unlabeled and labeled measured data to train the proposed models and improve their performance.

The rest of this paper is organized as follows: Section 2 describes a review of related works on the popular algorithms and learning strategies for the indoor localization and navigation task. In Section 3, we introduce the proposed method in detail, including the relevant models and a hybrid learning algorithm. Section 4 describes the details of our collected dataset, the experimental setting, results and analysis. Finally, a brief conclusion is provided in Section 5.

## 2. Related Work

For indoor localization and navigation, many works have shown the advantage of the ML algorithms to mitigate the fluctuation and noise of measured data from different complex environments. In the following, we review some recently popular algorithms and learning strategies used in the fingerprint-based positioning system.

K-nearest neighbor (KNN) is the simplest algorithm [20] used for fingerprint identification techniques [9,21,22], which selects the K-nearest annotated measurements from an offline radio map and calculates the average of their position coordinates to estimate the position of the target device. However, the KNN-based methods usually need to maintain such a large radio map to perform the frequent computation. Support vector machine (SVM) is another classical algorithm [23] used for classification problems. It could model linear and non-linear relationships among different fingerprint classes [10,24] by using the kernel mechanism for better generalization. Multi-layer perceptron (MLP) is an artificial neural network [25,26] consisting of multiple hidden layers that transform input data into target results in the output layer. It could be trained to extract robust fingerprint features [11,27] by using the back-propagation learning algorithm. If the amount of annotated data is limited in the training phase, the above models may suffer from the overfitting problem in the test phase. Some ensemble learning algorithms such as Adaboost [28] and random forest [29] are used to overcome this problem [12,13,30,31]. They train multiple weak learners (decision trees) by using the boosting or bagging method on the training set, and they integrate these learners into a strong learner for the final prediction. In recent years, deep learning (DL) has been increasingly used in the Wi-Fi-based positioning and navigation algorithms [14,15,16,17,18,19,32], such as DeepFi [32] and WiDeep [14], which adopt the channel state information (CSI) or the RSSI data from all subcarriers to train a neural network with more layers than the MLP network. Autoencoder (AE) is a popular network structure for feature extraction and is widely applied into these algorithms by developing different variants such as denoising AE. To achieve better performance, some recent works [15,19] use the stacked AE (SAE) structure to construct a deeper network, which will lead to higher computational cost in model deployment. The DL-based algorithms are also used to fuse multi-dimensional data [33,34] for improving the localization accuracy. Reinforcement learning [35] is a promising ML algorithm that learns a set of behaviors toward a defined goal based on the current state and environment. It could be used in the indoor navigation system [36,37,38,39] to provide an available path from the current position to an expected one.

Transfer learning (TL) [40] is an ML-based learning strategy that enables the ML algorithms to quickly adapt to the target domain by inheriting the learned knowledge from the source domain. The TL strategy could be applied in the fingerprint-based positioning system [41,42,43] to improve the model scalability and adaptability by using a small number of labeled data in the new environment to fine-tune the model weights trained in the previous environments. Semi-supervised learning [44] is an effective learning strategy that allows an ML model to learn the unlabeled data by using the pre-trained model to generate the associated simulated labels. The semi-supervised setup is well suited for the wireless data collection as it could be combined with other ML algorithms to reduce the annotation effort required for accurate positioning [8,36,45]. Unsupervised learning [46] is another common learning strategy used to reduce the dimensionality of input data by learning the compact and intrinsic features from unlabeled data. Some recent works [47,48,49] use the learning strategy to reduce the complexity and storage cost of fingerprint data.

## 3. Methodology

In this section, we provide a detailed description of our approach, including two deep models for the indoor localization and navigation problems, and a hybrid learning approach for the deep models with limited data.

### 3.1. Deep Encode–Decode Model for Indoor Localization Task

As shown in Figure 2, our models are mainly based on a classical neural network, called variational autoencoder (VAE) [50], which is used for unsupervised learning, especially in generative modeling tasks. The VAE network consists of two main components: an encoder and a decoder. The encoder first takes input data and maps it to a distribution over the latent space. The decoder then receives the sampled points from this latent space and reconstructs the input data. Instead of a deterministic mapping, VAE is designed to learn a probabilistic representation that follows a Gaussian distribution by having the encoder output the mean and variance of this distribution. This design can make the reconstruction more robust than the normal autoencoder because it introduces explicit regularization into the latent space to avoid possible overfitting in the generative process.

Inspired by the design of ResNet [51], which uses the skip connection to learn the residual function throughout the network, we apply densely fused connections to the VAE network and call it DF-VAE. Specifically, we first build an encoder consisting of four fully connected layers with 32, 64, 64 and 2 output channels. The encoder takes the RSSI data or the state data to generate different levels of intermediate representations. Then, we create a decoder with layers that are symmetric and connected to those in the encoder. During the decoding process, each layer performs a channel-wise sum of the current output and the corresponding one from the encoder. This design allows each layer in the decoder to learn a residual function with an identity mapping from the symmetric layer in the encoder, and it improves the gradient flow when training a much deeper network.

In our work, we formulate the indoor localization task as a position classification problem by creating a discrete grid map that contains specific rows and columns. Based on the DF-VAE network, we build a deep encode–decode localization model (DED-LM) by adding two new fully connected layers. The layers first encode the output of DF-VAE as a 64-dimensional feature representation and then generate a specific probability vector for the classification of the row or column numbers.

### 3.2. Deep Reinforcement Learning Model for Indoor Navigation Task

As shown in Figure 3, we model the indoor navigation as a Markov decision process (MDP) when given an instance in the indoor environment. The MDP is a mathematical framework used to model decision making in situations where outcomes are uncertain and can be influenced by the actions of a decision maker. The MDP consists of a set of states, a set of possible actions, a state transition function that describes the probability of moving from one state to another when an action is taken, and a reward function that assigns a numerical value to each state–action pair. The goal in the MDP is to find a policy that maximizes the expected cumulative reward over time.

In our work, we propose a deep reinforcement learning model based on the MDP framework. An agent is designed to contain a deep encode–decode action model (DED-AM), which consists of a DF-VAE network and two additional fully connected layers with 64 and 4 output channels. First, the agent receives the current state data including RSSI, position coordinate (row number and column number) and a distance from the target position, to generate a 4-dimensional score vector for the decision of four actions, i.e., move forward, move backward, move left, and move right. Then, the learning environment uses the state transition and reward functions to obtain the next state and a reward value, respectively, according to the action performed. During this process, the DED-AM model can be trained to choose an action with the highest reward each time, resulting in an optimal path from an initial position to a target one.

### 3.3. Hybrid Learning Approach for Deep Models with Limited Data

Given a dataset with labeled and unlabeled samples, we need to consider how to make the most of the data to train the deep models. The pseudo-code in Algorithm 1 shows a training overview of the proposed hybrid learning approach. The learning approach consists of four stages for different deep models with available data.

Firstly, we adopt all the RSSI data with or without labels to train the DF-VAE model. In this stage, we aim to use the unsupervised learning method to learn non-task-specific feature representations by reconstructing the input data. The reconstruction loss is formulated as follows:(1)$$L_{DF-VAE}=MSE\left(x,\hat{x}\right)+\lambda KL(p\left(z|x\right)||N\left(0,1\right)),$$
where $MSE(x,\hat{x})$ denotes the standard mean square error between the ground-truth input *x* and the reconstructed input $\hat{x}$. KL(p(z|x)||N(0,1)) denotes the the Kullback–Leibler (KL) divergence, which is used to measure the distance between the latent distribution p(z|x) and the standard normal distribution N(0,1). The weight λ is used to balance the contribution of MSE and KL.

In the second stage, we use the trained weights of the DF-VAE model to initialize the weights of the DED-LM model, and we iteratively refine them by using the semi-supervised learning method on the unlabeled RSSI data. Specifically, we first train a coarse DED-LM model by using the labeled data. Then, we apply the trained model with an adjustable score threshold to generate high-quality labels for the unlabeled data. Finally, we combine the new data with the existing labeled data to jointly fine-tune the model. By repeatedly fine-tuning the model, we can obtain a well-performing localization model for the classification of the row or column number in a grid map. The localization loss is defined as follows:(2)$$L_{DED-LM}=-\sum_{k=1}^{K}y_{k}log\left(p_{k}\right)$$
where $y_{k}$ denotes a ground-truth label or a pre-generated label of the *k*-th row or column, which is 1 if the sample is collected at that position and 0 otherwise. $p_{k}$ is the predicted probability that is obtained by applying the softmax function to the output of the model.

In the third stage, we first use the trained DED-LM model to predict the labels of all the unlabeled RSSI data and combine them with the labeled RSSI data. Then, we calculate the distances of all the data between the current position and a given target position. Finally, we train a new DF-VAE model for the reconstruction of the state information including RSSI, the current position and the distance. In the training process, we adopt the same reconstruction loss as $L_{DF-VAE}$ from the first stage.

In the last stage, we initialize the weights of the DED-AM model by using the trained weights of the DF-VAE model from the third stage. Given a sample from the RSSI data with the ground truth and predicted labels, we can obtain the current state information and perform an action to generate the next state information. The action is obtained by applying a ϵ-greedy strategy, in which a random action is chosen with a probability of ϵ or an action having a maximum reward is chosen with a probability of 1−ϵ. The strategy can achieve a trade-off between exploration and exploitation for action optimization. In our reinforcement learning algorithm, the reward function is designed as follows:(3)$$R_{t}=\left\{\begin{array}{cc} \frac{1}{\left|\right|C_{t}-T\left|\right|}*\left(w+h\right), & \mathrm{if}\;0<||C_{t}-T\left|\right|<\delta \\ -\left|\right|C_{t}-T\left|\right|/\left(w+h\right), & \mathrm{otherwise} \end{array}\right.$$
where $C_{t}$ and *T* denote the current position and the target position, respectively. The reward function has a positive value if the distance between $C_{t}$ and *T* is greater than 0 and less than a distance threshold δ. Otherwise, the DED-AM model receives a negative reward. *w* and *h* denote the number of rows and columns, which are used to scale the reward value. Based on the reward function, we formulate the action decision loss as follows:
**Algorithm 1:** Hybrid Learning Algorithm with Labeled and Unlabeled Data.**Input**: A dataset with labeled and unlabeled data $\left\{\left(X_{l},Y_{l}\right),X_{u}\right\}$, initialed environment, initialed network weights $w_{0}$, $w_{1}$, $w_{2}$ and $w_{3}$ of DF-VAE${}_{1}$, DF-VAE${}_{2}$, DED-LM and DED-AM.**Output**: The networks DED-LM and DED-AM.
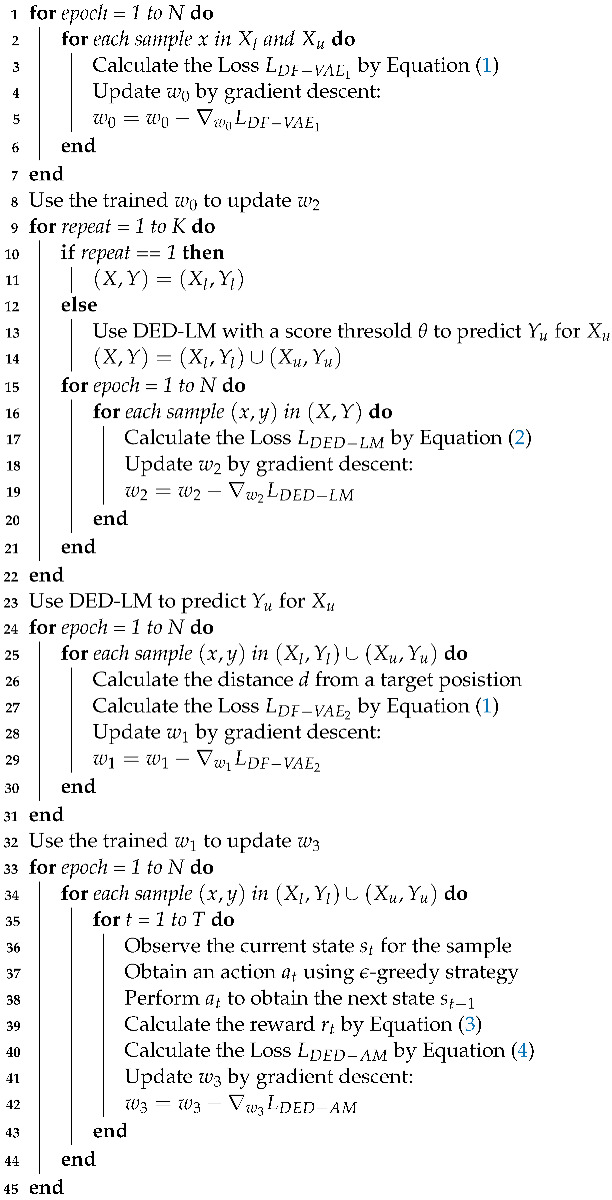

(4)$$L_{DED-AM}=MSE\left(R_{t}+\gamma \max f\left(S_{t+1}\right),f_{a_{t}}\left(S_{t}\right)\right)$$
where $f_{a_{t}}\left(S_{t}\right)$ denotes the score of the current action $a_{t}$ from the DED-AM model for the current state $S_{t}$. $\max f(S_{t+1})$ is the maximum score of all the actions from the DED-AM model for the next state $S_{t+1}$. The weight γ is used to adjust the radio of the current reward $R_{t}$ and the maximum score. The key idea of this loss is to make the model learn an expected score for a given action, in which the score is determined not only by an immediate reward but also by a delayed score used to evaluate the current direction.

## 4. Experiments

In this section, we provide a detailed description of the collected dataset and the experimental setting. The experimental results and relevant analysis are also provided by comparing popular algorithms with our method with different components.

### 4.1. Collected Dataset Using the Ackerman Robot

Our experiments are conducted in a 20 × 8 m exhibition hall on the first floor of the Hengqin–Macau youth entrepreneurship valley. As shown in Figure 4, we use the Ackerman robot with four wheels to automatically collect the Wi-Fi RSSI data and calculate the current position in this indoor environment. Figure 5 shows the robot’s active area divided into a grid map of 17 × 6 m, in which we randomly install seven Wi-Fi routers to simulate the indoor environments of some large stores with multiple Wi-Fi signals. In addition, we introduce two types of obstacles into the area, which are called Obstacle After and Obstacle Before. The former is added after the data collection and is common in some public indoor public places when new obstructions are added. The latter is present from the beginning, where the robot is not allowed to pass and collect the Wi-Fi signals.

Based on a pre-defined walking path, we make the robot collect seven RSSI values from different Wi-Fi routers 80 times at each passable position, in which only the collected data of the first 16 times are labeled with the corresponding position information, i.e., the row and column numbers in the grid map. As an example, we show the statistics of different RSSI values from two Wi-Fi routers in Figure 6, and the data follow an approximately normal distribution. Finally, we obtain 1584 labeled data and 6336 unlabeled data, which are used for the indoor localization and navigation tasks. In this paper, we refer to the robot-aided collected dataset as RA-CD.

### 4.2. Experiment Setting

All the experiments are performed on an Intel(R) Xeon(R) W-2140B CPU @ 3.20 GHz. The weights λ and γ in the loss functions $L_{DF-VAE}$ and $L_{DED-AM}$ are set to 0.0005 and 0.95, respectively. The score threshold θ is 0.2 and is used in the semi-supervised learning strategy. The probability threshold ϵ is set to 0.995 from the ϵ-greedy strategy in the reinforcement learning. The numbers of epochs *N*, repetitions *K* and times *T* from the proposed hybrid learning Algorithm 1 are set to 1000, 3 and 30, respectively. During the training, we use the Adam optimizer with an initial learning rate of 0.001.

### 4.3. Results on Wi-Fi-Based Indoor Localization Task

To validate the effectiveness of our method for indoor localization, we compare the proposed deep encode–decode model (DED-LM) with other popular algorithms on the RA-CD dataset. These algorithms include a classical SVM method using the Gaussian kernel function, the Adaboost and random forest algorithms using 50 decision trees as basic learners, an MLP neural network with two fully connected layers, and three popular deep learning networks used in the recent works [14,15,19], including denoising AE, SAE, and denoising SAE. To ensure a fair comparison, the AE structure in these networks is implemented using a similar structure to our model, but it is trained using a different learning strategy. SAE and denoising SAE consist of three stacked AE networks. In addition, we also perform a detailed comparison of our method variants to further evaluate the gain of each proposed component for our method. In the components, USL and SSL denote the unsupervised and semi-supervised learning strategies, respectively.

In this experiment, we first randomly divide the annotated dataset into a training set with 1267 samples and a test set with 317 samples. Then, we train all the methods for the classification of the row or column numbers and evaluate their accuracies on the test set. In order to observe the stability of the performance, we execute the process five times using different random seeds and report the average accuracy with the standard deviation. As shown in Table 1, our method achieves a better performance than other methods with an obvious margin for the accuracy and the stability. We also note that denoising AE can improve the performance of MLP by using a deeper network structure with an appropriate learning strategy. By stacking multiple AE networks, SAE and denoising SAE have a further improvement in accuracy, but they inevitably introduce more computational cost. In addition, the results of the AdaBoost algorithm are impressive and show a promising direction for exploiting the ensemble learning strategy. From Table 2, we show the effect of different components on the both accuracy and stability. VAE + MLP has a dramatically degraded performance by adding a VAE network as the backbone of MLP. It means that a deeper network is not always effective if it cannot be trained well. By introducing the dense skip connections in VAE, DF-VAE + MLP achieves a significant performance improvement as the gradient flow is improved when training the network. Moreover, we employ two additional learning strategies USL and SSL to make full use of the unlabeled data during training. The results show that our hybrid learning strategy can achieve a better and more stable performance for the row and column localization. Figure 7 demonstrates the variation of the training losses of our method variants and other deep learning methods with the increasing epochs. We find that a deeper network has a more unstable loss and easily converges to a local optimum, such as VAE + MLP. By using the proposed learning strategies, we observe that although the loss has an increased amplitude, its oscillation frequency is inhibited with a better convergence result.

### 4.4. Results on Wi-Fi-Based Indoor Navigation Task

In this subsection, we focus on evaluating the performance of the proposed deep reinforcement learning model for the indoor navigation task on the RA-CD dataset. The previous experiment has shown the advantage of the designed network structure in DF-VAE + MLP. Therefore, we mainly make a comparison of our method variants using the proposed DED-AM model and different learning strategies. In contrast to the localization task, the navigation performance is assessed by observing the estimated movement path from an initial position to a target position.

In this experiment, we train three navigation models for all the variants given three target positions, i.e., (17, 3), (1, 4) and (9, 6), as shown in Figure 5. Then, we estimate all the movement paths from each non-obstacle position to the given target position by setting the maximum step to 25, and we report the average distance between each end position of all the estimated paths and each target position as well as the average reward during model inference. From Figure 8, we observe that the DF-VAE + MLP method without any learning strategies shows an unstable navigation performance, as the average reward and distance for the first target position are much lower and longer than those for the other target positions, respectively. By adding the USL learning strategy during training, the navigation performance is obviously improved, but there is still a not-insignificant gap between the metrics of the first target position and the others. It means that the weight initialization from the USL strategy can help the model learn a better local solution in reinforcement learning, but it cannot ensure a stable performance for all the cases. In contrast, the SSL strategy achieves a significant improvement in performance whether in terms of the average metrics or their stability for all the target positions. This suggests that the pseudo-labels of unlabeled data from the SSL strategy can effectively improve the model optimization process by introducing additional task-specific knowledge. By combining the USL and SSL strategies, our model can achieve not only the competitive or higher average reward than using the SSL strategy alone but also the shortest average distance that is more important to users for the indoor navigation task.

## 5. Conclusions

In this paper, we present a robot-aided hybrid deep learning approach that includes a dataset automatically collected by a robot, two deep learning models for Wi-Fi-based indoor localization and navigation, and a novel hybrid learning strategy used to train the models. Experimental results show the performance advantage of our approach when compared to recent popular algorithms. Although the proposed deep encode–decode model is intuitive and effective for the localization task, it introduces a degree of capacity redundancy by solving the column and row localization separately. In addition, our model currently only considers the single RSSI information from a limited exhibition hall and still has a potential improvement by using new Wi-Fi data such as the CSI data.

In future work, we will explore how to extend our model to support multi-task decision making for indoor localization and multi-modal learning from different types of Wi-Fi data. Furthermore, we will consider how to construct a more challenging benchmark by applying our method to more indoor environments.

## Figures and Tables

**Figure 1 sensors-23-06320-f001:**
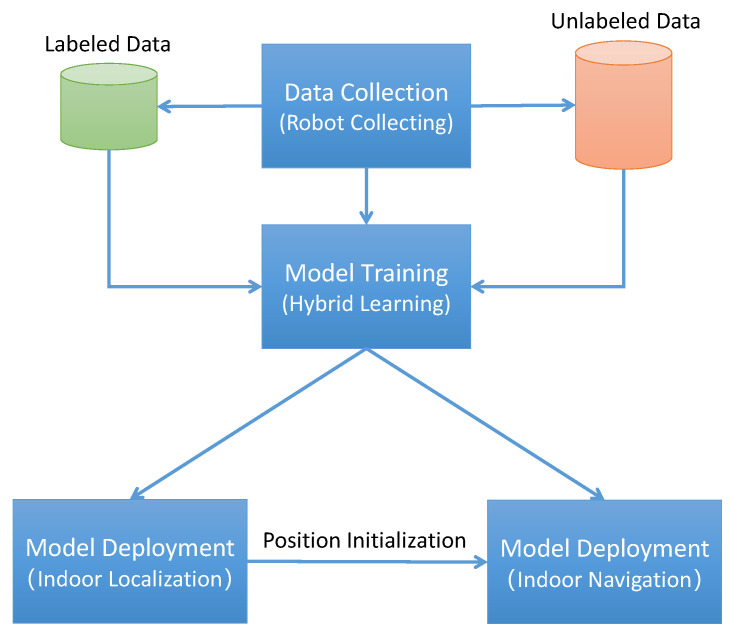
Workflow of the proposed approach for indoor localization and navigation.

**Figure 2 sensors-23-06320-f002:**
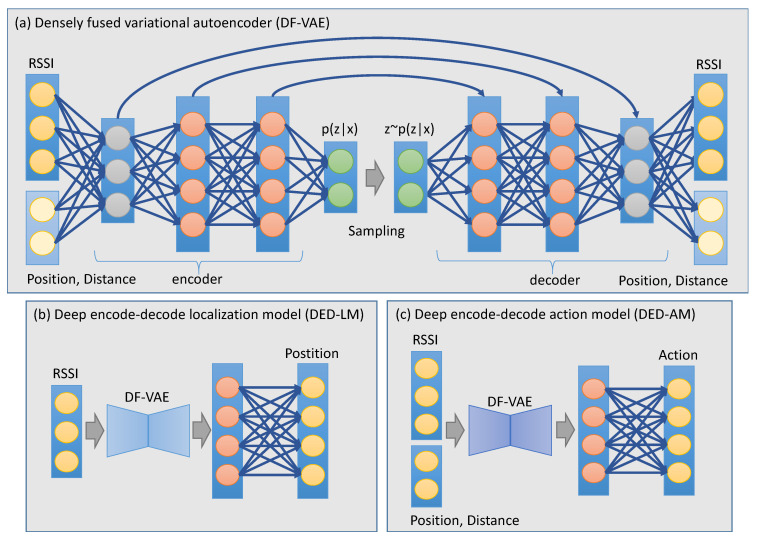
Overall structures of a densely fused variational autoencoder (DF-VAE) for reconstruction learning (**a**) and two deep encode–decode models (DED-LM and DED-AM) for position classification (**b**) and action decision (**c**), respectively.

**Figure 3 sensors-23-06320-f003:**
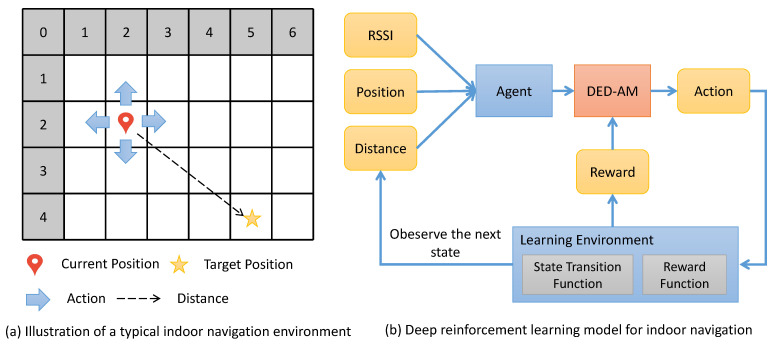
Illustration of a typical indoor navigation environment (**a**) and the process of deep reinforcement learning for an indoor navigation task (**b**).

**Figure 4 sensors-23-06320-f004:**
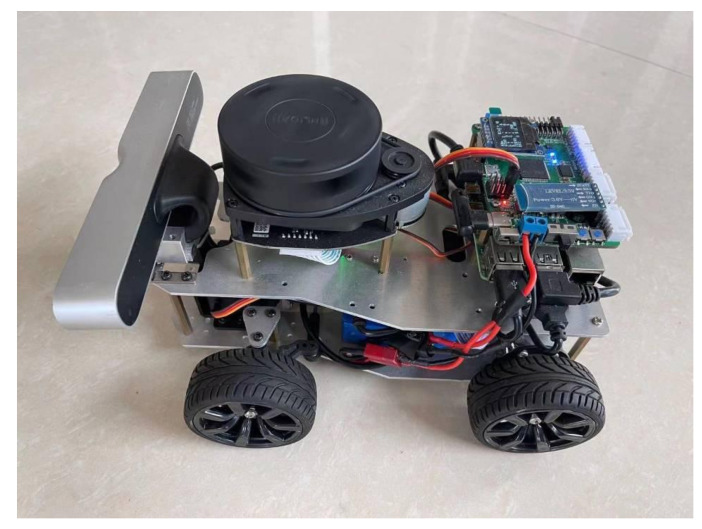
Illustration of the Ackerman robot for indoor RSSI data collection.

**Figure 5 sensors-23-06320-f005:**
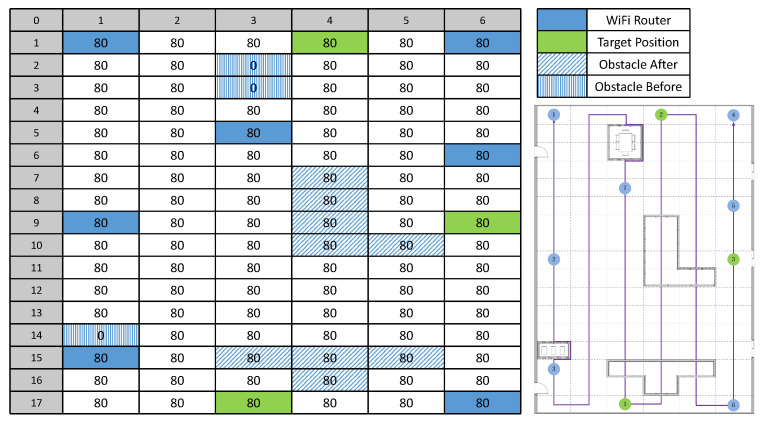
Overview of an indoor space with all Wi-Fi routers, target positions, obstacles and a pre-defined walking path for data collection. We use the Ackerman robot to collect the RSSI data from 7 different Wi-Fi routers in a 17 × 6 grid space with two types of obstacles before and after the collection. The number in each cell indicates the number of collections at that position.

**Figure 6 sensors-23-06320-f006:**
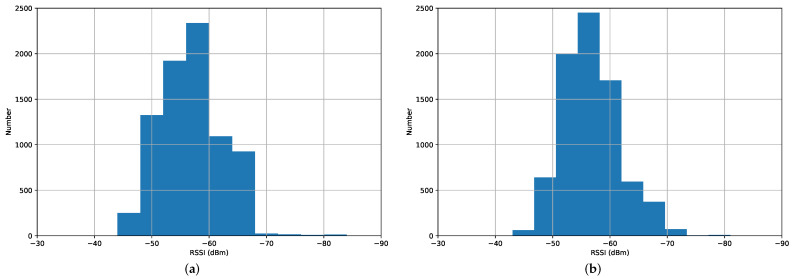
Statistics of different RSSI values from two Wi-Fi routers (**a**,**b**).

**Figure 7 sensors-23-06320-f007:**
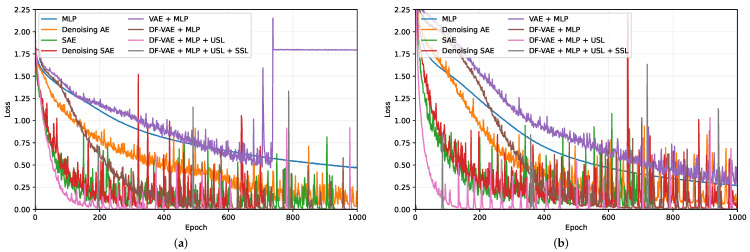
Comparison of the loss curve of other methods and the proposed variants for the column (**a**) and row (**b**) localization on the training set of the RA-CD dataset.

**Figure 8 sensors-23-06320-f008:**
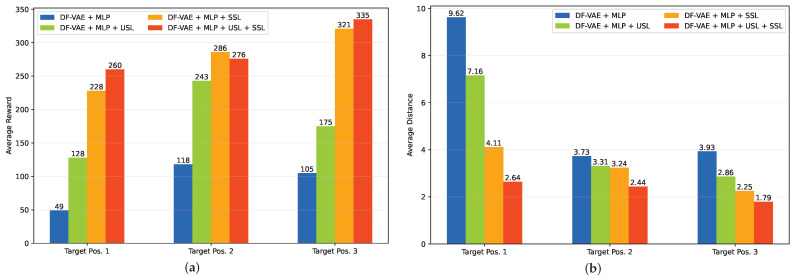
Comparison of the average reward (**a**) and the average distance (**b**) of our method variants using different learning strategies on the RA-CD dataset.

**Table 1 sensors-23-06320-t001:** Comparison of the accuracy of popular methods on the test set of the RA-CD dataset.

Method	Column Accuracy	Row Accuracy	Mean Accuracy
SVM	67.70% ± 1.01%	75.46% ± 1.70%	71.58% ± 1.05%
Random Forest	71.23% ± 2.31%	76.40% ± 4.61%	73.82% ± 2.93%
MLP	80.19% ± 2.85%	86.56% ± 2.32%	83.38% ± 2.37%
AdaBoost	90.91% ± 1.85%	93.12% ± 0.97%	92.02% ± 1.05%
Denoising AE	88.90% ± 3.49%	82.78% ± 0.99%	85.84% ± 2.05%
SAE	94.89% ± 0.50%	94.26% ± 1.47%	94.57% ± 0.72%
Denoising SAE	95.27% ± 1.00%	92.81% ± 0.86%	94.04% ± 0.76%
Ours	96.03% ± 0.91%	95.96% ± 0.76%	95.99% ± 0.51%

**Table 2 sensors-23-06320-t002:** Comparison of the accuracy of our method variants on the test set of the RA-CD dataset.

Method	Column Accuracy	Row Accuracy	Mean Accuracy
MLP	80.19% ± 2.85%	86.56% ± 2.32%	83.38% ± 2.37%
VAE + MLP	59.43% ± 15.63%	72.93% ± 3.90%	66.18% ± 7.99%
DF-VAE + MLP	90.28% ± 3.30%	92.30% ± 0.65%	91.29% ± 1.92%
DF-VAE + MLP + USL	95.71% ± 0.71%	95.46% ± 0.93%	95.58% ± 0.73%
DF-VAE + MLP + USL + SSL	96.03% ± 0.91%	95.96% ± 0.76%	95.99% ± 0.51%

## Data Availability

The data is available at https://github.com/MUST-AI-Lab/RA-HDL (accessed on 2 June 2023).

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
