# Peer review of "Wi-Fi-Based Indoor Localization and Navigation: A Robot-Aided Hybrid Deep Learning Approach"

_sensors, 2023, doi:10.3390/s23146320_

Round 1

Reviewer 1 Report

The manuscript entitled "Wi-Fi-based Indoor Localization and Navigation: A Robot-Aided Hybrid Deep Learning Approach" proposed a robot-aided data collection strategy to obtain limited but high-quality labeled data and a large amount of unlabeled data. Furthermore, design two deep learning models based on variational autoencoders for the localization and navigation tasks, respectively. To make full use of the collected data, a hybrid learning approach is developed to train the models by combining supervised, unsupervised, and semi-supervised learning strategies. Extensive experiments show that our method achieves promising localization and navigation performance in a complex indoor environment with obstacles and that the proposed components are effective in providing incremental improvements. However, I have a few major concerns:

1) The motivation or research gaps have not been highlighted in a comprehensive way which makes the proposed system compromising. 
2) Collecting datasets via robots is cost effecting task, also may be less confident, please defend it with logical justification. 
3) why only these methods have been used to develop a hybrid approach, please justify it.
4) The results before and after hybridization should be explored so that we can have a better explanation and a justification for why the need for the hybridization method occurred.
5) comparison with previous studies should also be added to prove the validity of the proposed work.

Reviewer 2 Report

The authors propose a localization scheme for indoor environments based on two deep-learning models. Through robot-aided data collection, they obtain high-quality labeled data while their models are trained through supervised, unsupervised, and semi-supervised learning strategies. 

The article at hand is generally well written, without grammatical issues. The novelty of the work is clear, as presented in the methodology section, while it is evaluated through an extensive experimental protocol. 

Author Response

Thanks very much for your comments.

Reviewer 3 Report

A robot-aided data collection technique and also two deep learning models based on variational autoencoder along with a hybrid learning method to train the models for localization and navigation purposes in indoor areas, have been proposed in this paper. The authors have also provided sufficient experimental results to show the promising performances of their proposed method compared to other existing schemes. However, the paper has several flaws for which the following suggestions are given to further improve the technical merit of this paper.

1. Replace ‘are also need’ at line 42 with “are also needed’.

2. The article does not provide literature review of recently proposed machine learning based Wi-Fi fingerprint localization systems. So, recently proposed machine learning based Wi-Fi fingerprint positioning methods that addresses various other issues of fingerprint positioning such as positioning overhead, storage overhead, etc., should be briefly discussed in Section 2.

3. Various notations, variables & symbols used to describe the proposed method must be explicitly defined in a table at the beginning of Section 3.

4. An architectural framework for the proposed localization & navigation system showing its different stages like data collection, model training, model development and finally location estimation etc., and the relationships between them should be provided for better understanding of how this proposed localization system works.  

5. A schematic diagram of the experimental site showing Wi-Fi routers, data collection points or predefined walking path, target position, obstacles etc. should be provided in Section 4.  

6. The proposed localization & navigation system should be evaluated in terms of positioning error or accuracy.

Reviewer 4 Report

This paper proposes a robot-assisted data collection strategy to obtain limited but high-quality labeled data and large amounts of unlabeled data. Two deep learning models based on variational autoencoder are designed for localization and navigation tasks. In order to make full use of the collected data, a hybrid learning method is proposed, which combines supervised learning, unsupervised learning and semi-supervised learning strategies to train the model. The topic is interesting and the presentation is clear. The author compared their proposed method with traditional methods. It would be more convincing if the results could be compared with the latest literature.

Minor editing of English language required.

Author Response

Thanks very much for your comment. We have added new comparative experiments and relevant analysis of the recent works in Section 4. In addition, we have carefully checked and corrected the grammatical errors and tried our best to improve some inappropriate descriptions in the new manuscript.

Reviewer 5 Report

This paper provides two different deep learning algorithms for indoor robot localization and navigation based on Wi-Fi technology. Additionally, this paper proposes an easy data collection strategy for machine learning algorithms training.  On the positive side, the article is presented in a clear and well-structured form. The addition of the pseudo-code on page 6, which describes the developed algorithm, also improves the replicability of the study. However, it would have been useful to provide more information and make some changes to various aspects of the developed localization and navigation algorithm.

1. In recent years, Wi-Fi fingerprint-based indoor-localization method has been widely investigated (it is possible to find many publications in 2022 and 2023 about this technology in the literature). Therefore, the state-of-the-art should collect and describe more similar works on the use of Wi-Fi for the location and navigation of indoor robots. Furthermore, this paper should outline how it differs from these similar works, how the proposed work improves on the current state-of-the-art (ease of implementation, performance, etc.), and why is this current review still relevant and of interest to the scientific community.

2. The abstract section should indicate the main conclusions or interpretations. The limitations of the work should also be highlighted in the conclusions.

3. 21/47 references are not recent publications (within the last 5 years). Try (if possible) to update references to more recent ones.

4. Would it be possible to visualize what the data collected by the robot that feeds the DL algorithm mentioned in section 4.1 look like?

5. Some grammatical errors and typos could be corrected.

6. The data availability statement indicates that GitHub has been accessed on 1 September 2023. How is this possible? Moreover, the GitHub page does not exist.

This paper provides two different deep learning algorithms for indoor robot localization and navigation based on Wi-Fi technology. Additionally, this paper proposes an easy data collection strategy for machine learning algorithms training.  On the positive side, the article is presented in a clear and well-structured form. The addition of the pseudo-code on page 6, which describes the developed algorithm, also improves the replicability of the study. However, it would have been useful to provide more information and make some changes to various aspects of the developed localization and navigation algorithm.

1. In recent years, Wi-Fi fingerprint-based indoor-localization method has been widely investigated (it is possible to find many publications in 2022 and 2023 about this technology in the literature). Therefore, the state-of-the-art should collect and describe more similar works on the use of Wi-Fi for the location and navigation of indoor robots. Furthermore, this paper should outline how it differs from these similar works, how the proposed work improves on the current state-of-the-art (ease of implementation, performance, etc.), and why is this current review still relevant and of interest to the scientific community.

2. The abstract section should indicate the main conclusions or interpretations. The limitations of the work should also be highlighted in the conclusions.

3. 21/47 references are not recent publications (within the last 5 years). Try (if possible) to update references to more recent ones.

4. Would it be possible to visualize what the data collected by the robot that feeds the DL algorithm mentioned in section 4.1 look like?

5. Some grammatical errors and typos could be corrected.

6. The data availability statement indicates that GitHub has been accessed on 1 September 2023. How is this possible? Moreover, the GitHub page does not exist.

Round 2

Reviewer 1 Report

The authors adequately replied to all concerns and made significant changes in the updated manuscript. I would like to accept this manuscript for publication. 

Minor grammatical errors and the use of more appropriate verbs are required.

Author Response

Thank you for the comment. We have carefully checked and corrected some  grammatical errors and inappropriate descriptions in the new manuscript.

Reviewer 2 Report

Congratulations to the authors. The manuscript exceeds my requirements and is ready for publication.

Author Response

Thanks very much for your comment.